# Patient-Derived Xenograft Models of Breast Cancer and Their Application

**DOI:** 10.3390/cells8060621

**Published:** 2019-06-20

**Authors:** Takahiko Murayama, Noriko Gotoh

**Affiliations:** Division of Cancer Cell Biology, Cancer Research Institute, Kanazawa University, Kanazawa 920-1192, Japan; takahiko_rsmk@yahoo.co.jp

**Keywords:** PDX, breast cancer, humanized mice

## Abstract

Recently, patient-derived xenograft (PDX) models of many types of tumors including breast cancer have emerged as a powerful tool for predicting drug efficacy and for understanding tumor characteristics. PDXs are established by the direct transfer of human tumors into highly immunodeficient mice and then maintained by passaging from mouse to mouse. The ability of PDX models to maintain the original features of patient tumors and to reflect drug sensitivity has greatly improved both basic and clinical study outcomes. However, current PDX models cannot completely predict drug efficacy because they do not recapitulate the tumor microenvironment of origin, a failure which puts emphasis on the necessity for the development of the next generation PDX models. In this article, we summarize the advantages and limitations of current PDX models and discuss the future directions of this field.

## 1. Introduction

Breast cancer is the leading cause of death in women worldwide, though much progress has been made in diagnosis and treatment strategies over the several decades. Tumors, including those of breast cancer, are composed of highly heterogeneous populations [1]; this heterogeneity is the major cause of the difficulty in eradicating them with current therapies. The inter-tumor heterogeneity of breast cancer also makes it difficult to accurately predict drug efficacy with actively used cancer models like cell lines cultured in vitro and in vivo. Patient-derived xenograft (PDX) models have recently been developed to better reflect the heterogeneity of patient tumors of origin. These models are expected to improve therapeutic strategies against breast cancer.

In PDX models, cancer cells or small tumor tissues derived from patients are injected into immune-deficient mice. The PDX models are mainly used to assess the efficiency of anti-tumor drugs or to clarify the characteristics of cancer cells and their microenvironment since the models closely resemble the original tumors of patients. PDX models have been established for many types of tumors including breast cancer [2], colorectal cancer [3], pancreatic cancer [4], B cell lymphoma [5], lung cancer [6], and ovarian cancer [7]. These PDX models have started to be widely used for drug development and pre- or co-clinical trials. Though current PDX models have some limitations and drawbacks, they have contributed to progress in basic and clinical cancer research. In this review, we compare the advantages of PDX and other cancer models, summarize research utilizing PDX models, and discuss some limitations and future directions of PDX models, mainly focusing on breast cancer PDXs.

## 2. Generation of PDX Models

### 2.1. Immunodeficient Mice

To establish PDX models, patient-derived tumors have to be injected into highly immunodeficient mice, because the mouse immune system could eradicate transplanted cancer cells and prohibit tumor engraftment. Nude mice or severe combined immunodeficiency (SCID) mice are rarely used for the establishment of PDX models, although they are often used to establish cell line xenograft models. Non-obese diabetic (NOD)-scid, NOD.Cg-Prkdc^scid^Il2rg^tm1Wjl^/SzJl (NSG), or NOD.Cg-Prkdc^scid^Il2rg^tm1Sug^/ShiJic (NOG) mice, which are characterized by a relatively high immunodeficiency due to a decrease or complete lack of natural killer (NK) cell functions, are the major tools for PDX establishment [8,9]. Furthermore, NSG and NOG mice can be a recipient of human hematopoietic stem cells [10]. As described later, immunodeficient mice engrafted with human immune systems have been established as a powerful tool for the next generation of PDX models [11], as well as for infectious disease mouse models [12].

### 2.2. Patient-Derived Tumors

Tumor samples obtained from patients by surgical resection or biopsy should be maintained in sterile conditions to prevent bacterial contamination. Before implantation, tumors are cut into small pieces and/or digested into single cells [13]. The tissues or the cells can be implanted heterotopically (in most cases subcutaneously) or orthotopically into immunodeficient mice. Heterotopic injection is often chosen because the method is much easier, and the monitoring of tumor size can be done accurately for any type of tumor. However, for the breast cancer PDX, most researchers choose orthotopic implantation into mammary fat pads [14,15,16,17], as it is not technically difficult and can be accurately monitored. In addition, orthotopic implantation has many advantages compared to heterotopic implantation. The microenvironment around the implanted tumor is more closely preserved, which enables tumors to interact with microenvironment components. Orthotopic implantation also increases the incidence of metastases [16], leading to a wide application of PDX models.

## 3. Current Representative Line of Cancer Models

In order to enable researchers to do cancer research efficiently, many advances have been made in the establishment of cancer models. Each model has several advantages and limitations. Therefore, choosing an appropriate model for their own purpose is very important. The advantages and limitations of four cancer models—cell line (in vitro), cell line xenograft, genetically engineered mouse, and PDX—are summarized below and in Table 1.

### 3.1. Cell line (cultured in vitro)

Cancer cell lines are the most frequently used tool for cancer research [18]. They can be maintained inexpensively and treated very easily, so many researchers use this model for the first screening of newly developed drug candidates, for investigating cancer characteristics, and for many other purposes. However, cancer cell lines cultured in vitro completely lack interaction with the tumor microenvironment, which is the main reason for the difficulties of predicting drug efficacy and in understanding drug resistance mechanisms with this model [19]. Taking these characteristics into consideration, this model should be applied to basic studies and very early stages of drug development in which the large number of samples are needed.

### 3.2. Cell Line Xenograft Model

To recapitulate the interaction between cancer cells and the tumor microenvironment, cancer cell lines are transplanted into immunodeficient mice. These cell line xenograft models have been widely used since the first development of the soft tissue sarcoma cell line xenograft model [20]. If cancer cells are transplanted orthotopically, the tumor microenvironment resembles the condition in which the original tumor existed in the patient. Cell line xenograft models are often used for obtaining proof of concept in vivo in the relatively early stages of drug development. In addition, this model is frequently used to understand cancer genetics and drug resistance mechanisms. The popularity of cell line xenograft models is due to their high availability, their lesser costs compared to PDX models, and their high take rates. However, cell line xenografts have limited predictive value for drug efficacy [21].

The main factor for this disadvantage is thought to be the change of the cell properties between the original tumor and established cancer cell lines. Cancer cell lines tend to lose the heterogeneous characteristics of original tumors by the selective pressure on cell culture in vitro [22]. In addition, the long term culture with a culture medium can alter the properties of cancer cell lines little by little. Another factor which makes cell line xenograft models inappropriate for drug efficacy prediction is that most cell lines are derived from highly aggressive malignant tumors. This tendency makes it difficult to recapitulate inter-tumor heterogeneity. Therefore, cell line xenograft models are not the best tools for precise medicine and co-clinical trials.

### 3.3. Genetically Engineered Mouse Model

Genetically engineered mouse models of breast cancer are established by several approaches [23,24]. One approach is the introduction of an exogenous gene of interest (oncogene) by directly injecting DNA into a fertilized egg [25]. Another approach is the knockout of the gene of interest (tumor suppressor gene) by introducing a targeting vector-encoding modified version of the tumor suppressor gene into mouse embryonic stem cells. The development of a Cre/loxP system enabled the conditional knock in and knock out in specific organs, like mammary gland, which made genetically engineering systems more convenient tools.

This type of mouse model has often been used for understanding of the process of tumor initiation, early development, and relapse after therapies [26]. Spontaneous tumor initiation and relapse occur within the appropriate microenvironment in this model. Therefore, this model has been widely used when investigating how a specific gene could contribute to tumor initiation and relapse. For example, by using this mouse model, Goel et al. showed that cyclin D1/Cyclin-dependent kinase 4 (CDK4) mediated resistance to human epidermal growth factor 2 (HER2)-targeted therapy and that CDK4/6 inhibitors delayed a HER2 positive tumor relapse [27]. However, genetically engineered mice are not always thought to be very suitable for pre- or co-clinical studies. This model cannot completely mimic the human tumors because cancer cells are originally derived from mouse cells. Historical models of mouse mammary tumor virus (MMTV)-infected tumors do not histologically show human breast tumor characteristics [28]. Even if driver genes are introduced into the cells and breast tumors are formed, the heterogeneity of human tumors is not completely recapitulated in these models. Furthermore, it takes very long time (~1year or more) to establish the model. After interesting gene mutations are found in tumor patients, it is impossible to establish the new mouse model with such mutations for the process of precision medicine.

### 3.4. PDX Model

PDX models have strong advantages in the field of pre- and co-clinical studies when compared with cell line xenografts and genetically engineered mouse models. It has been shown that tumors formed in PDX models resemble the original tumors in patients, both histologically and genetically. Furthermore, Zhang et al. clearly proved that PDX tumors show comparable treatment responses to those observed clinically. In their study, 12 of 13 PDX tumors from 13 patients treated with the same drug showed a matched response with the corresponding clinical response [2].

The recent development of highly immunodeficient mice has enabled us to establish PDX models. Despite some disadvantages, including low take rates and high costs, PDX models are becoming more and more popular, especially for the use of drug efficacy prediction. Recently, The Jackson Laboratory (Bar Harbor, Maine, USA) started to distribute PDX models, and many consortiums on PDX have been established, which has made PDX models more accessible for researchers worldwide.

PDX models have also been used for metastatic models [15]. Since modeling tumor metastasis is difficult for genetically engineered mice, PDX metastasis models have the potential to make great progress in metastatic research. However, there are still some challenges to be overcome. The common metastatic sites of breast cancer PDX models are only lymph nodes and lungs, even though breast cancer in patients also frequently metastasizes to the brain and bones [29]. Therefore, an improved strategy for metastatic PDX establishment will be needed to better predict metastatic behavior by PDX models.

## 4. PDX Models of Each Breast Cancer Subtype

Breast tumors are normally classified into four subtypes to decide the therapeutic strategies—luminal A, luminal B, and human epidermal growth factor 2 (HER2) positive and triple negative [30,31]. Luminal A and luminal B subtype tumors express an estrogen receptor (ER) and/or a progesterone receptor (PR), and most of them are dependent on estrogen for their growth. HER2 positive tumors are driven to grow by the activation of signaling pathways regulated by HER2 homodimerization and heterodimerization with other HER family members—HER1, HER3, and HER4. Triple negative subtypes have the worst prognosis and show no therapeutic targets at the moment. As therapeutic strategies differ among these subtypes, the development of all types of PDX models is needed [2,32,33,34,35,36,37,38,39,40,41,42,43,44,45,46,47,48,49,50,51,52,53,54,55,56,57,58,59,60,61,62,63,64,65,66,67] (Table 2).

### 4.1. Luminal A and Luminal B Subtypes

Most tumors of these subtypes are efficiently treated by therapeutic strategies targeting ER and estrogen production [68], suggesting that they are strongly dependent on the function of estrogen. Therefore, estrogen pellets are often injected into immunodeficient mice before tumor transplantation when producing luminal-type PDX models [2,50,69]. Kabos et al. showed that the xenografts derived from luminal tumor expressed ER at a similar level to patient tumors of origin [50]. This indicated that the dependency on estrogen did not change after transplantation. Another report using luminal subtype PDX models for pre-clinical purpose made it clear that the B-cell lymphoma 2 (Bcl-2) Homology 3 (BH3) mimetic improved the tumor response to the antiestrogen tamoxifen [70].

### 4.2. HER2 Positive Subtype

HER2 positive subtype tumors show relatively low take rates, and the number of therapeutic studies using this type of PDX is limited. However, Kang et al. successfully utilized this model and showed that WW-binding protein 2 (WBP2) helped the inhibitory effect of trastuzumab, a monoclonal antibody targeting HER271. They suggested that WBP2 would be useful as a companion diagnostic for the management of HER2 positive breast cancer with trastuzumab-based therapies. In addition, PDX models of HER2 positive breast cancer-derived brain metastases were also developed by Ni et al. [71]. By using these PDX models, they found that the combined inhibition of phosphoinositide 3-kinase (PI3K) and mammalian target of rapamycin (mTOR) led to the durable regression of metastasized tumors. The brain-metastasized models of this type of tumor also resembled the parental metastasized tumors of patients histologically, highlighting the ability of PDX models to maintain original features.

### 4.3. Triple Negative Subtype

This type of breast tumors does not express ER, PR, or HER2. Therefore, they cannot be treated with endocrine therapy or anti-HER2 targeting strategies. The triple negative subtype represents about 15–20% of breast cancer patients and shows the worst prognosis of the major four subtypes due to the lack of effective treatment other than cytotoxic chemotherapy [72,73]. Therefore, the development of new therapeutic strategies is urgently needed. PDX models of triple negative breast cancers are often utilized for such a purpose [66,74,75]. One reason for their popularity is that, in all the subtypes of breast cancer, the triple negative type shows the highest take rate, partly because of its strong aggressiveness [76,77].

Research using a triple negative breast PDX by EI Ayachi et al. showed that the inhibition of Wnt/β-catenin/HMGA2/EZH2 signaling deprived chemo-resistance to doxorubicin in this type of tumor [78]. Based on the results, they proposed Wnt signaling network targeting therapy as a promising strategy for triple negative breast cancers. Furthermore, a pre-clinical rational for developing treatment approaches using the Notch1 monoclonal antibody [79], Wee1 kinase inhibitor [80], Wnt inhibitor [78], and so on, has been established by utilizing these PDX models.

## 5. Application of PDX Models for Clinical Use

PDX models are superior to cell line xenografts and genetically engineered mouse models, especially in the field of pre- and co-clinical studies, because they have much higher predictive values [81,82,83]. As long as PDXs are maintained in vivo by directly passaging from mouse to mouse, their character closely resembles that of their parental tumors for several generations. Though the take rate of first transplantation from patient to mouse varies dependent on tumor types [77,84,85,86,87], the second and subsequent take rates from mouse (or frozen stock) to mouse are generally high. Therefore, if PDXs are successfully established by first transplantation, researchers can use them for many purposes by passaging formed tumors to a larger number of mice.

### 5.1. PDX Models for Drug Development

A very large number of novel drug candidates drops in phase Ⅱ of clinical trials, which is the cause of the enormous costs needed for drug development. The major reason for this problem is the poor predictive values of current cancer models (cell line xenograft or genetically engineered mouse) used in pre-clinical studies. PDX models have the potency to improve such a situation by enabling researchers to predict drug efficacy more precisely [88], as suggested by Zhang et al. [2]. Karamboulas et al. established a large collection of PDX models of head and neck squamous cell carcinoma (HNSCC). They showed the efficacy of CDK4 and CDK6 inhibitors for HNSCC with CCND1 and CDKN2A genomic alterations [89,90]. For breast cancer, Grunewald et al. evaluated the activity of a novel fibroblast growth factor receptor (FGFR) inhibitor, rogaratinib, using cancer cell lines and breast cancer PDX models [91]. They found that the inhibitor has a strong efficacy for FGFR overexpressing cells, both in vitro and in vivo. Furthermore, based on these findings, they started clinical trials of rogaratinib for patients with FGFR overexpressing tumors.

### 5.2. PDX Models for Precision Medicine

Making appropriate therapeutic regimes based on the features of each tumor leads to better treatment responses. Breast cancers are categorized based on the expression levels of ER, PR, and HER2, and many patients have benefitted from the therapeutic strategies developed according to these categorizations. On the other hand, patients with triple negative subtype tumors show a worse prognosis due to the lack of the good therapeutic targets [92]. In order to improve this situation, researchers are trying to further divide triple negative types into some detailed subtypes, such as basal-like 1, basal-like 2, mesenchymal, and luminal androgen receptor [93,94]. PDX models in drug screening tests will accelerate these studies by enabling us to predict drug efficacy on each triple negative tumor subtype.

### 5.3. PDX Models for Co-Clinical Trials

In order to determine appropriate therapeutic strategies for each patient, co-clinical trials using mouse models are parallelly operated with clinical treatments [95,96]. Originally, genetically engineered mouse models that had the similar genetic abnormalities to patients were used for this purpose [97,98,99]. However, to predict drug efficacy more precisely, PDX models are being used more these days. For some tumor types, including ovarian cancer and head and neck sarcomas, studies to confirm the efficacy of PDX co-clinical trials are now ongoing. Though some reports, including one by Julic et al., support this concept, there are many hurdles that must be crossed to use breast cancer PDX models for co-clinical trials. Firstly, the take rates of breast cancer are very low, which makes breast cancer PDX models unreliable for therapeutic options. Secondly, it takes a long time (3 months–1 year) to establish the model, which could cause delays to determining therapeutic strategies.

## 6. Limitations of Current PDX Models

### 6.1. Lack of Immune Cells

In many types of tumors, including breast cancer, immune cells in tumor microenvironments play very important roles for tumor growth and progression [100,101]. However, current PDX models are established by transplanting tumors into highly immunodeficient mice, which lack the majority of an immune system required in order to obtain higher take rates. Therefore, PDX models cannot reproduce the interaction between cancer cells (or other microenvironment components) and immune cells which exist in patient tumors. This may make it difficult to completely predict drug efficacy and to understand drug resistance mechanisms [102,103]. Fortunately, as described later, next generation PDX models have been established to overcome this kind of limitation.

### 6.2. Low Take Rates

The take rates of transplanted tumors greatly differ among tumor types of origin. In general, the take rates of patient derived breast cancers are very low (approximately 10–25% on average) [21,77], although the development of pre-exposure methods of estrogen has slightly enhanced the take rates of luminal type tumors. The low take rates and long term incubation periods in transplanted mice make it difficult for us to utilize breast PDX models for pre- or co-clinical studies. If further studies develop more suitable mice for PDX models or better methods of tumor transplantation, which contribute to higher rates of breast tumors, breast cancer PDX models will be more popular in clinical studies.

### 6.3. High Cost

The financial aspect should also be taken into consideration, because highly immunodeficient mice are very expensive. Maintaining those mice in a clean environment also takes a high cost, as it takes long time before tumors are engrafted and begin to grow in PDX models.

In the case of co-clinical trials using PDX models, a whole genome analysis of original tumors will also be needed. There are still some problems to be solved before PDX models show their true value in clinical settings.

## 7. Next Generation PDX Models with Human Immune System

It has become clear in the last few decades that components of the tumor microenvironment play very important roles for cancer cell growth and maintenance [104]. The tumor microenvironment is composed of very heterogeneous populations: Cancer cells, cancer associated fibroblasts (CAFs), vascular epithelial cells, many kinds of immune cells, and platelets [105]. In order to reproduce more accurate conditions of tumor origins, much progress has been made in PDX establishment methods. One example is to transplant patient-derived CAFs or mesenchymal stem cells (MSCs), along with cancer cells, to recapitulate the interaction between cancer cells and CAFs [106].

Very recently, humanized mice have started to gather attention as an attractive tool for PDX models [61,64,107] (Figure 1). To enhance the take rates of patient derived tumors, highly immunodeficient mice have been widely used for PDX model establishment. However, mice lacking an immune system cannot recapitulate the interaction between cancer cells and immune cells in the tumor microenvironment. To overcome this disadvantage, mice engrafted with a human immune system are expected to be a promising tool for the next generation PDX models. In 2008, Pearson et al. suggested a protocol for the generation of humanized mice with human immune cells [108]. Adult immune deficient mice like NSG mice are irradiated by 240 cGy whole body gamma irradiation. After four hours, T-cell depleted hematopoietic stem cells (HSCs) containing CD34+ cells are injected into the lateral tail vein. Then, human HSCs are engrafted in the immune deficient mice 10 to 12 weeks after injection [108]. Rosato et al. established triple negative breast cancer PDX models with these humanized mice [61] and provided the evidence supporting the use of humanized PDX models as good models for the pre-clinical investigation of immune-based therapies. In addition to immune-based therapies, humanized PDX models will also improve the predictive values for other types of therapeutic strategies.

## 8. Conclusions

In this review, we discuss the advantages and limitations of current PDX models, usage examples of PDX models derived from all major subtypes of breast cancers, and the development of next generation PDX models with human immunity. Concrete evidence has accumulated to suggest that breast cancer PDX models are valuable tools for predicting drug efficacy in pre- and co-clinical trials because these models well-maintain the heterogeneity and properties of the patient tumors of origin. However, there is a large engraftment bias toward the triple negative subtype. We still have to improve the take rates, particularly of luminal A, B, and HER2 positive subtype tumors, to make good use of PDXs for clinical purposes.

Though current PDX models have some limitations, including the loss of immune systems, researchers are now developing next generation models to overcome such drawbacks. If PDX models, which more accurately reflect the features of the human tumor microenvironment, become popular, cancer research—both in basic and clinical levels—will be highly accelerated. In order to utilize breast cancer PDX models in clinical settings, low take rates and the high cost of PDX establishment are big problems that must be overcome. International collaborative networks may work together to solve these problems. While there are still some obstacles, PDX models have a great potential to improve therapeutic strategies against breast cancer.

## Figures and Tables

**Figure 1 cells-08-00621-f001:**
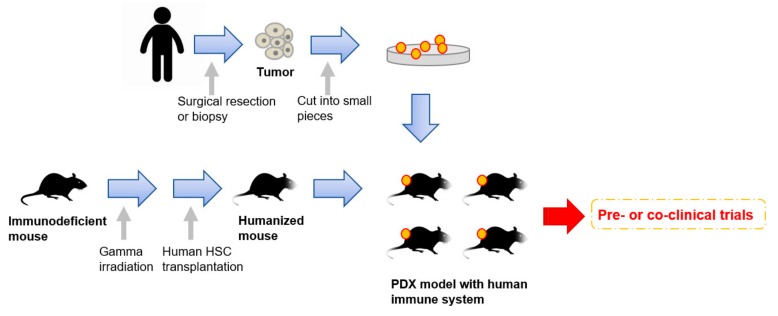
An overall procedure for the generation of PDX models engrafted with a human immune system. Humanized mice are generated by a human hematopoietic stem cells (HSC) transplantation into irradiated immunodeficient mice. Patient-derived tumors obtained by surgical resection or biopsy are sliced into small pieces and then transplanted into the humanized mice.

**Table 1 cells-08-00621-t001:** Advantages and limitations of each cancer model.

	Advantages	Limitations	Recommendations to Overcome Limitations
Cell line (cultured in vitro)	·Maintained inexpensively ·Treated very easily ·Grow infinitely	·Completely lack the tumor microenvironment ·Can’t maintain original cell properties ⇒Very low predictive value	·Should be used in basic studies and very early stages of drug development ·Co-culture with cancer associated fibroblasts (CAFs) or immune cells will improve the predictive value
Cell line xenograft	·High take rates ·Slightly recapitulate tumor microenvironment ·Take short time to be established	·Can’t reproduce heterogeneity ·Can’t maintain the original cell properties ⇒Low predictive value	·Should be used in the relatively early stages of drug development with a large number of mice, which can reflect the inter-tumor heterogeneity
Genetically engineered mouse	·Recapitulate tumor initiation and early development process ·Gene of interest can be studied in detail ·Can be increased easily after establishment	·Can’t reproduce heterogeneity of human tumor ⇒Low predictive value ·Take long time to be established	·Should be used when investigating how a specific gene of interest could contribute to tumor initiation and relapse
PDX	·Partly recapitulate tumor microenvironment ·Maintain histologic and genetic features of origin ⇒High predictive value ·Can be used for metastatic model	·Low take rate ·Very expensive ·Take long time to be established	·Development of new immunodeficient mice and/or better methods of tumor transplantation will improve the take rates and the cost

**Table 2 cells-08-00621-t002:** Applications of each breast cancer subtype patient-derived xenograft (PDXs).

Authors	Host Mouse	Tissue Source	Subtypes of PDX (Number)	Method	Reference
Agnoletto and collegues	nude	primary, metastasis	triple negative (7), HER2+ (2), luminal (2)	interscapular	[31]
Al-Hajj and collegues	NOD/SCID	primary	triple negative (4)	orthotopic	[32]
Arango and collegues	nude	primary	triple negative (5)	orthotopic	[33]
Bruna and collegues	NSG	primary, biopsy, plueral effusion	triple negative (24), HER2+ (6), luminal (52)	orthotopic	[34]
Castroviejo-Bermejo and collegues	NSG	primary, biopsy	triple negative (8), luminal (5)	orthotopic	[35]
Contreras-Zárate and collegues	NSG	metastasis	triple negative (3), HER2+ (5), luminal (1)	orthotopic	[36]
Coussy and collegues	nude	primary	triple negative (61)	orthotopic	[37]
Cruz and colleagues	nude	primary	triple negative (9), luminal (1)	orthotopic	[38]
Dávila-González and collegues	SCID/Bg	primary	triple negative (5)	orthotopic	[39]
DeRose and collegues	NOD/SCID	primary, pleural effusion	triple negative (5), HER2+ (2), luminal (5)	orthotopic	[40]
Evans and collegues	NOD/SCID, nude	primary	triple negative (24)	orthotopic	[41]
Fatima and collegues	NSG	primary	triple negative (2)	orthotopic	[42]
Fleming and collegues	NOD/SCID	pleural effusion	-	orthotopic	[43]
Formisano and collegues	SCID/Bg	primary	luminal (2)	orthotopic	[44]
González-González and collegues	NSG	primary	triple negative (2)	orthotopic	[45]
Hsu and collegues	NSG	primary	luminal (2)	orthotopic	[46]
Hu and collegues	NOD/SCID	primary	-	orthotopic	[47]
Jung and collegues	NOD/SCID	primary	triple negative (24)	orthotopic	[48]
Kabos and collegues	NOD/SCID, NSG	primary, metastasis	triple negative (2), luminal (8)	orthotopic	[49]
Kanaya and collegues	NSG	primary	luminal (9)	orthotopic	[50]
Li and collegues	NOD/SCID	primary, metastasis	triple negative (12), HER2+ (2), luminal (8)	orthotopic	[51]
Liu and collegues	NSG	pleural effusion	HER2+ (2), luminal (2)	orthotopic	[52]
Ma and collegues	NOD/SCID	primary, metastasis	triple negative (3)	orthotopic	[53]
Marangoni and collegues	nude	primary	triple negative (15), HER2+ (2), luminal (1)	orthotopic	[54]
Matossian and collegues	SCID/Bg	primary	triple negative (1)	orthotopic	[55]
Méndez-Pertuz and collegues	nude	primary, metastasis	luminal (7)	orthotopic, lower flank	[56]
Merino and collegues	NSG	primary	triple negative (2)	orthotopic	[57]
Pillai and collegues	NOD/SCID	primary	triple negative (3), luminal (2)	orthotopic	[58]
Rather and collegues	NSG	primary	triple negative (1)	s.c. in the right flank	[59]
Rosato and collegues	NSG		triple negative (5)	orthotopic	[60]
Ruiz de Garibay and collegues	nude	primary	triple negative (1)	orthotopic	[61]
Ryu and collegues	NSG, NOG	primary	triple negative (9), HER2+ (7), luminal (4)	orthotopic	[62]
Wang and collegues	SCID/Bg	primary	triple negative (2)	orthotopic	[63]
Wang and collegues	nude	primary	luminal (1)	orthotopic	[64]
Zhang and collegues	SCID/Bg, NSG	primary, pleural effusion	triple negative (12), HER2+ (3), luminal (2)	orthotopic	[2]
Zhang and collegues	NOD/SCID	primary, metastasis	triple negative (7)	orthotopic	[65]
Zhang and collegues	NSG	biopsy	-	orthotopic	[66]

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
