# Peer review of "Patient-Derived Xenograft Models of Breast Cancer and Their Application"

_cells, 2019, doi:10.3390/cells8060621_

Round 1

Reviewer 1 Report

The author, in this paper, reviewed materials on limitations and benefits of patient-derived xenograft models, which have been established to date, particularly in relation to breast cancer research. This reviewer would like to inform you of some concerns that should be supplemented by the authors. 1. In an investigation of HER different types, the explanation with the triple negative or HER2 + only could be considered a deficient content, since the tissue of patients with breast cancer tends to have very different dependencies on HER1, 2, and 3, respectively.  2. There is very little description of clinical outcome or implication related to drug screening using various models and prediction of prognosis after treatment. It may be necessary to complement the above two points.

Author Response

Reviewer #1

1.     In an investigation of HER different types, the explanation with the triple negative or HER2 + only could be considered a deficient content, since the tissue of patients with breast cancer tends to have very different dependencies on HER1, 2, and 3, respectively.

⇒ We categorized breast cancers according to the major subtype definition (luminal A or B, HER2 and triple negative). However, as reviewer #1 pointed out, HER1, 3 and 4 play important roles to activate HER2 regulated signaling pathways. Therefore, we added the explanation on HER2-related signals in lines 176- 177.

2.     There is very little description of clinical outcome or implication related to drug screening using various models and prediction of prognosis after treatment.

⇒ The concrete data showing high predictive value of PDX models were lacking in the previous version of manuscript. Therefore, according to the suggestion, we introduced a study by Zhang et al. in lines 148-151, which showed the high predictive value of this model. In addition, we made a little change in lines 239-244, in which we stated poor predictive values of cell line xenograft and genetically engineered mouse models that leads to a high drop rate of drug candidates in clinical trials.

We look forward to hearing from you regarding our submission. We would be glad to respond to any further questions and comments that you may have.

Reviewer 2 Report

In this article, the authors review the patient-derived xenograft (PDX) models of tumors with focus on breast tumor. The article is well written, and comprehensively described. The strong aspect is the detailed discussion on available ‘’patient-derived xenograft models’’ and how they are being implemented experimentally. However, authors should give a way forward to overcome limitations in existing models, e.g. those limitations listed in tables.

1.       While authors mention the advantages and limitations of different models, in tables, I strongly recommend authors to add an additional column to provide recommendations or alternatives over those limitations.  

2.       Since authors discuss both the available models and their application, the later part has been less described or less focused. I suggest authors to emphasize the applications part.

3.       Figure 1. Define HSC in legends.  

Author Response

Reviewer #2

1.     While authors mention the advantages and limitations of different models, in tables, I strongly recommend authors to add an additional column to provide recommendations or alternatives over those limitations.

⇒ According the suggestion by reviewer #2, we added a column to Table 1. for providing recommendations to overcome limitations of current mouse models (between lines 82 and 83).

2.     Since authors discuss both the available models and their application, the later part has been less described or less focused. I suggest authors to emphasize the applications part.

⇒ Exactly as reviewer #2 pointed out, applications of current models (cell lines cultured in vitro, cell line xenografts and genetically engineered mice) were less described compared to the introduction of available models. Therefore, we added some explanations on applications in lines 91-94 (cell lines cultured in vitro), lines 102-105 (cell line xenografts) and lines 128-134 (genetically engineered mice).

3.     Figure 1. Define HSC in legends.

⇒ In legend of Figure 1, we defined the HSC as hematopoietic stem cells.

We look forward to hearing from you regarding our submission. We would be glad to respond to any further questions and comments that you may have.

Round 2

Reviewer 1 Report

All concerns that were raised by this reviewer have been well addressed. There is no additional issue to raise.